# TDM-Based Approach for Properly Managing Intravenous Isavuconazole Treatment in a Complex Case Mix of Critically Ill Patients

**DOI:** 10.3390/antibiotics14080777

**Published:** 2025-08-01

**Authors:** Milo Gatti, Matteo Rinaldi, Riccardo De Paola, Antonio Siniscalchi, Tommaso Tonetti, Pierluigi Viale, Federico Pea

**Affiliations:** 1Department of Medical and Surgical Sciences, Alma Mater Studiorum University of Bologna, 40138 Bologna, Italy; matteo.rinaldi23@unibo.it (M.R.); tommaso.tonetti@unibo.it (T.T.); pierluigi.viale@unibo.it (P.V.); federico.pea@unibo.it (F.P.); 2Clinical Pharmacology Unit, Department for Integrated Infectious Risk Management, IRCCS Azienda Ospedaliero-Universitaria di Bologna, 40138 Bologna, Italy; 3Infectious Disease Unit, Department for Integrated Infectious Risk Management, IRCCS Azienda Ospedaliero-Universitaria di Bologna, 40138 Bologna, Italy; 4Specialisation School of Clinical Pharmacology and Toxicology, Alma Mater Studiorum Università di Bologna, 40138 Bologna, Italy; riccardo.depaola@studio.unibo.it; 5Anesthesia and Intensive Care Medicine, IRCCS Azienda Ospedaliero-Universitaria di Bologna, 40138 Bologna, Italy; antonio.siniscalchi@aosp.bo.it; 6Division of Anesthesiology, Department of Anesthesia and Intensive Care, IRCCS Azienda Ospedaliero-Universitaria di Bologna, 40138 Bologna, Italy

**Keywords:** isavuconazole, real-time TDM-guided approach, critically ill patients, invasive fungal infection, overexposure, underexposure, hepatic test disturbance

## Abstract

**Objectives:** To assess the role of a real-time therapeutic drug monitoring (TDM)-guided expert clinical pharmacological advice (ECPA) program of isavuconazole in preventing under- or overexposure with the intent of improving efficacy and safety outcomes in the critically ill patients. **Methods:** This retrospective study included critical patients receiving intravenous isavuconazole for prophylaxis or treatment of invasive fungal infections (IFI) and undergoing at least one TDM-guided ECPA in the period 1 March 2021–31 March 2025. Desired isavuconazole exposure was defined as trough concentrations (C_min_) of 1.0–5.1 mg/L. Efficacy outcome was assessed by means of bronchoalveolar (BAL) galactomannan (GM) index, breakthrough IFI, and 30-day mortality rate, whereas safety was assessed by means of hepatic test disturbances (HTD). Univariate analysis was carried out for assessing potential variables associated with isavuconazole under- or overexposure and for comparing features of solid organ transplant (SOT) recipients vs. non-SOT patients. Proportions of isavuconazole C_min_ underexposure, desired exposure, and overexposure were assessed at different timepoints from starting therapy. Trends over time of HTD in relation to isavuconazole exposure were assessed separately in patients having HTD or not at baseline. **Results:** Overall, 32 critical patients were included. A total of 166 TDM-guided ECPAs were provided. Median (IQR) average isavuconazole C_min_ was 3.5 mg/L (2.1–4.6 mg/L). Proportions of ECPAs with isavuconazole C_min_ under- and overexposure were 4.2% (7/166) and 16.3% (27/166), respectively. Patients experiencing underexposure had higher body mass index (30.1 vs. 25.5 kg/m^2^; *p* < 0.001). Trends of isavuconazole C_min_ under- and overexposure changed over time, significantly decreasing the former (10.5% <7 days vs. 4.3% 7–28 days vs. 0.0% >28 days; *p* < 0.001) and increasing the latter (5.3% <7 days vs. 12.8% 7–28 days vs. 29.3% >28 days; *p* < 0.001). HTD occurred in 15/32 patients, most of whom (10/15) were affected just at baseline. Patients with transient or persistent overexposure trended toward a higher risk of HTD compared to those without (33.3% vs. 8.3%; *p* = 0.11). **Conclusions:** A real-time TDM-guided approach could be a valuable tool for optimizing isavuconazole exposure, especially whenever dealing with obese patients or with prolonged treatment.

## 1. Introduction

Invasive fungal infections (IFIs) represent a major disease burden among critically ill patients, since more than 65% of them may be affected with remarkable mortality [1,2,3]. Although usually considered as opportunistic infections affecting immunocompromised patients, in the last decade, an ever-growing increasing prevalence of IFIs has been reported also among non-neutropenic critically ill patients admitted to the intensive care unit (ICU) [1,2]. Particularly, SARS-CoV-2 infection, influenza, and solid organ transplant were shown to be major underlying conditions increasing the risk of mold infections caused by *Aspergillus* spp. or Mucorales among critically ill patients [3].

Voriconazole and isavuconazole are triazole antifungals considered as first-line treatment of mold infections [4,5]. Voriconazole is indicated for invasive aspergillosis but, unfortunately, its use may be burdened by some specific issues, namely a high potential of drug–drug interactions and a dose-dependent hepato- and/or neuro-toxicity risk [6,7,8]. Additionally, voriconazole pharmacokinetics may be affected by the genetic polymorphism of CYP2C19, namely the major isoform involved in its biotransformation. Consequently, the wide interindividual pharmacokinetic variability makes plasma exposure unpredictable under standard daily doses so that, nowadays, therapeutic drug monitoring (TDM) of voriconazole is considered mandatory [8].

Isavuconazole is a valuable alternative to voriconazole in the management of invasive aspergillosis and also to liposomal amphotericin B in that of mucormycosis [9,10]. Isavuconazole represents the active metabolite of the water-soluble prodrug isavuconazonium sulfate [11]. It is a moderately lipophilic drug (LogP 3.46), with low molecular weight (437.47 Da), high plasma protein binding (98–99%), a large volume of distribution (308–542 L), and extensive biotransformation by CPY3A4 [11]. Compared to voriconazole, isavuconazole has a mild to moderate potential of drug–drug interaction and a lower hepatotoxicity risk [6,11,12]. From a pharmacodynamic perspective, isavuconazole exhibits a broad-spectrum antifungal activity similar to those of voriconazole against *Candida* spp. and *Aspergillus* spp. but with the advantage of showing a valuable activity also against *Mucorales* spp. [11]. In regard to *Candida* spp., isavuconazole showed favorable MIC_50_ and MIC_90_, ranging from ≤0.03 to 0.5 mg/L, with the exception of *Candida auris*, for which higher MIC_50_ and MIC_90_ have been reported (up to 1 mg/L) [11]. Furthermore, isavuconazole showed in vitro excellent activity against *Aspergillus* spp., including strains with reduced susceptibility or developing resistance to voriconazole, exhibiting MIC_50_ and MIC_90_ values ranging between 0.5 and 4 mg/L [11]. Additionally, isavuconazole showed also good in vitro activity against some Mucorales species, particularly *Rhizopus* spp. with an MIC_50_ of 1–2 mg/L [11]. The pharmacokinetics of isavuconazole are quite predictable, and TDM is not considered as strictly needed [6,11,12]. However, some recent experiences suggested that, in critically ill patients, underexposure to isavuconazole during treatment with standard dosages might occur more frequently than in non-critically ill patients, with a prevalence ranging from 12.1% to 34.4% [7,13,14,15].

The aim of this study was to assess the role of a TDM-based approach for properly managing efficacy and/or safety of intravenous isavuconazole treatment in a highly com-plex case mix of critically ill patients.

## 2. Results

Overall, a total of 32 patients were included whose demographic and clinical features are summarized in Table 1. Median (IQR) age was 62 (51–65) years, with a male preponderance (78.1%). Median (IQR) BMI was 25.5 (22.3–27.8) kg/m^2^, with five patients being obese (15.6%). SOT (21 cases; 65.6%) was the most prevalent underlying disease, followed by hematological malignancies (6 cases; 18.8%) and by CAPA (5 cases; 15.6%).

Median (IQR) SOFA score at starting isavuconazole treatment was 8 (5–11) points. Among the included patients, 19/32 (59.4%) required vasopressors support, 25/32 (78.1%) needed mechanical ventilation, 16/32 (50.0%) underwent CRRT, and 2/32 (6.3%) underwent hemoadsorption with cytosorb. No patient underwent ECMO. A total of 22 out of 32 patients had moderate or severe hypoalbuminemia (68.7%).

IFIs were proven, probable, and possible in 1 (3.1%), 25 (78.1%), and 2 cases (6.3%), respectively, whereas the remaining 4 patients (12.5%) received antifungal prophylaxis in the absence of any IFI diagnosis.

Among the 28 patients with IFI, GM index was always positive on BAL with a median (IQR) of 2.6 (2.0–4.2). In three cases (9.4%), GM index was simultaneously positive also in serum with a median (IQR) value of 2.9 (2.0–7.4). Positive BAL culture for *Aspergillus* isolate was documented in six cases (*Aspergillus flavus* in two cases and *Aspergillus fumigatus*, *Aspergillus nidulans*, *Aspergillus terreus*, and *Aspergillus fumigatus* + *Aspergillus nidulans* in one case each). Isavuconazole was first-line therapy in 6/28 cases and second-line in the other 22/28 (namely, 9 after breakthrough IFI occurring under liposomal amphotericin B or anidulafungin treatment, 7 due to toxicity concerns and/or failure in attaining optimal voriconazole exposure, and 6 as a switch after liposomal amphotericin B therapy in the absence of any breakthrough IFI or toxicity).

Among patients receiving isavuconazole as prophylaxis, it was used as first-line in two cases and second-line in the other two (due to hepatotoxicity during posaconazole or voriconazole prophylaxis).

Median (IQR) isavuconazole treatment duration was 35 (12.5–47.75) days. Overall, isavuconazole average C_min_ during treatment course were within, below, or over the desired range in 29 (90.6%), 2 (6.3%), and 1 patient (3.1%), respectively. One patient had co-treatment with a strong CYP3A4 inducer (phenytoin), whereas none received strong CYP3A4 inhibitors and/or P-gp modulators. In the former, isavuconazole underexposure persisted even after increasing dosage up to 400 mg/day. Conversely, in a heart transplant recipient, isavuconazole overexposure persisted even after decreasing dosage to 100 mg every 48 h. Overall, a total of 166 TDM-guided ECPAs were provided, with a median (IQR) number per patient of 4 (2–7). Median (IQR) isavuconazole average C_min_ was 3.5 mg/L (2.1–4.6 mg/L), and median (IQR) daily MD was 200 mg (200–200 mg). Isavuconazole C_min_ were within, below, and over the desired range in 131 (78.9%), 7 (4.2%), and 28 (16.9%) of the 166 instances, respectively. Isavuconazole dosing adjustments were recommended in 14 out of 166 TDM-guided ECPAs (8.4%), these being decreases and increases in 8/14 (57.1%) and 6/14 (42.9%), respectively.

No breakthrough IFI occurred. In four patients (12.5%), antifungal therapy was switched to liposomal amphotericin B (in 3/4 as a re-switch after second-line treatment with isavuconazole), but all of these had clinical failure and passed away. Among the 23 patients having follow-up assessment of BAL GM index over time, 16 (69.6%) had ≥50% decrease from baseline within 14 days. The 30-day mortality rate was 31.3%.

Univariate analysis investigating potential variables associated with isavuconazole underexposure is reported in Table 2. Overall, median BMI > 30 kg/m^2^ was the only factor with significant association (30.1 vs. 25.5 kg/m^2^; *p* < 0.001), whereas coadministration of strong CYP3A4 inducers trended toward significant association (50.0% vs. 0.0%; *p* = 0.06). Univariate analysis investigating potential variables associated with isavuconazole overexposure was unfeasible due to low-frequency events.

Overall, HTD occurred in 15/32 patients, most of whom (10) just had altered baseline hepatic biochemical parameters. Specifically, HTD occurred in 9/21 patients always having the desired isavuconazole exposure (7 having and 2 not having altered baseline hepatic biochemical parameters) (Figure 1) and in 6/9 patients having transient (Figure 2A, *n* = 5) or persistent (Figure 2B, *n* = 1) isavuconazole overexposure (3 having and 3 not having altered baseline hepatic biochemical parameters). No HTD occurred in the two patients having transient (*n* = 1) or persistent (*n* = 1) isavuconazole underexposure. No patients required isavuconazole discontinuation due to HTD occurrence. Among the 10 patients having baseline alterations of hepatic biochemical parameters, severity of HTD was of grade 1, 2, or 3 by CTCAE in 5 (50.0%), 3 (30.0%), and 2 cases (20.0%), respectively. Among the five patients having normal baseline hepatic biochemical parameters, it was of grade 1 or 2 in two cases each (40.0%) and of grade 4 in one case (20.0%). This latter patient developed severe multiorgan failure and subsequently passed away due to severe CAPA. Patients having transient or persistent overexposure trended toward a higher risk of HTD compared to those not having it (33.3% vs. 8.3%; *p* = 0.11).

Trends of isavuconazole exposure over time in all of the included patients (*n* = 32) are shown in Figure 3 and summarized in Table 3.

Overall, trends of isavuconazole C_min_ under- and overexposure changed over time, significantly decreasing the former (10.5% <7 days vs. 4.3% 7–28 days vs. 0.0% >28 days; *p* < 0.001) and increasing the latter (5.3% <7 days vs. 12.8% 7–28 days vs. 29.3% >28 days; *p* < 0.001).

Univariate analysis comparing features in SOT vs. non-SOT patients is summarized in Table 4. Overall, the two groups did not differ in terms of demographics and of underlying pathophysiological conditions. Non-SOT patients received isavuconazole more often for prophylaxis (36.4% vs. 0.0%; *p* = 0.009) and had shorter median treatment duration (13 vs. 42 days; *p* = 0.02), lower median isavuconazole average C_min_ (2.20 mg/L vs. 3.44 mg/L; *p* = 0.006), more C_min_ values below the desired range (17.2% vs. 1.5%; *p* = 0.002), and less above it (0.0% vs. 19.7%; *p* = 0.005).

## 3. Discussion

Our study may provide further knowledge about the role that a real-time TDM-guided ECPA strategy may have in managing efficacy and/or safety of intravenous isavuconazole treatment in critically ill patients. Overall, this approach turned out to be effective either in minimizing the proportion of cases having isavuconazole under- or overexposure (<10% of cases had C_min_ outside of the desired range) or in limiting the occurrence and/or the severity of HTD among a highly complex case mix of ICU patients in terms of underlying diseases and of risk of multiorgan failure.

Interestingly, the rate of isavuconazole underexposure observed in our study was two- to five-fold lower (C_min_ < 1 mg/L in around 5% of cases, mainly during the first 7 days of treatment) than observed previously in ICU patients [7,13,14,15], with a median (IQR) isavuconazole C_min_ of 3.5 (2.1–4.6) mg/L. Most of these occurred in the first week of treatment. Mikulska et al. found that the rate of isavuconazole C_min_ < 1 mg/L among 33 ICU patients was of 12.1% and that the mean (±SD) isavuconazole C_min_ was significantly lower compared with that of 39 non-ICU patients (2.02 ± 1.22 vs. 4.15 ± 2.31 mg/L; *p* < 0.001) [15]. Höhl et al. found that the rate of isavuconazole C_min_ < 1 mg/L among 41 critically ill patients was of 31.7%, with a median value of 1.74 mg/L [14]. Bertram et al. found that the rate of isavuconazole C_min_ < 1 mg/L among 62 patients was of 34.4%, with a median C_min_ of 1.64 mg/L [13]. Finally, Roberts et al. reported that the rate of isavuconazole C_min_ < 1 mg/L among 10 ICU patients was of 40% [7].

Importantly, in our study, elevated BMI was the only factor significantly associated with isavuconazole underexposure. This (BMI > 25 kg/m^2^) is in agreement with what was previously observed in other studies carried out in the ICU setting [13,14,15]. Specifically, in a population PK study conducted among 18 critically ill patients receiving isavuconazole treatment for CAPA, high median BMI (i.e., 29.2 kg/m^2^) was associated with an increased volume of distribution (i.e., 850 L) [18]. Noteworthy, elevated BMIs, by increasing isavuconazole volume of distribution, may lead to underexposure during standard dosing regimen, as previously reported [14,15,18]. Since BMI is expected to affect mainly the adequacy of the loading dose, this may explain why underexposures were mainly observed at the beginning of treatment (10% of overall TDM during the first week). In this regard, it is worth noting that a physiologically based PK model of informed precision dosing optimization showed that increasing the LD of oral isavuconazole up to 300 mg q8 h for 48 h could be a valuable approach for properly dealing with this issue in the obese population [19]. Anyway, adopting a TDM-guided approach immediately after ending the loading period, as we usually did, could be complementary in promptly correcting eventual persisting underexposure. In regard to the maintenance period, isavuconazole underexposure could occur during co-treatment with strong inducers of CYP3A4, which may accelerate isavuconazole clearance [11]. Unfortunately, the only occurrence (co-treatment with phenytoin) we had in our study did not have the statistical power to demonstrate this hypothesis.

Conversely, the proportion of patients experiencing isavuconazole overexposure during the maintenance period increased over time, reaching around 30% after 28 days from starting treatment. This is consistent with what was previously reported in the available evidence [20,21,22,23] and may be explained by the very long elimination half-life (around 130 h), which may become even much longer in those ICU patients developing multiorgan failure [24,25]. In this regard, a retrospective study including 19 hematological patients receiving prolonged isavuconazole therapy found that isavuconazole C_min_ increased linearly by 0.032 mg/L for each day of treatment [20] and that higher C_min_ were independently associated in multivariate analysis with length of treatment (*p* < 0.001) [20]. A population pharmacokinetic model conducted among 50 patients with IFI showed that the probability of isavuconazole overexposure increased progressively over time, ranging from 11.6% at day 7 to 27.7% at day 28, up to 39.2 at day 60 [21]. Importantly, in our study, the proportion of overexposure increased from 5.3% during week 1 to 12.8% in weeks 1–4 up to 29.3% in week >4 and, overall, was higher in SOT vs. non-SOT critically ill patients. Consistently, the TDM-guided approach may be very valuable when dealing with long-lasting treatment, especially in SOT recipients, as this may prevent overexposure potentially leading to HTD.

The current literature suggests an upper threshold for isavuconazole C_min_ of 5.1 mg/L as being safe. Specifically, in a retrospective cohort of 19 non-critical oncohematological patients, an isavuconazole C_min_ of 5.13 mg/L was associated with gastrointestinal adverse events, mainly nausea and/or anorexia/hyporexia, but no specific threshold for HTD was identified [20]. Interestingly, by applying this potential toxicity threshold in our cohort, we found that HTD occurrence trended toward a higher risk among patients experiencing transient or persistent isavuconazole overexposure. In the vast majority of cases, HTD occurred after at least 20 days of treatment, with previous evidence consistently supporting emergence of adverse events in prolonged isavuconazole treatment [20,26,27]. Noteworthy, among patients experiencing transient isavuconazole overexposure, a reversal from HTD to baseline conditions was observed once C_min_ moved to the desired range thanks to TDM-guided dosing adjustments. An increase in serum AST/ALT levels was observed in more than half of our cases having HTD, other than that of cholestatic indexes as previously reported [25,27,28]. This finding is consistent with a recent retrospective study including 95 outpatients receiving prolonged isavuconazole treatment, in which transaminitis occurred in 29.7% of cases [29]. Notably, the authors identified an isavuconazole C_min_ of 5.86 mg/L as being the best threshold for this adverse event occurrence [29]. Anyway, it must be recognized that, to date, the role that several confounding factors related to the complexity of case mix (i.e., underlying diseases, concomitant drugs, and occurrence of multiorgan failure) might have in the emergence of HTD could not be ruled out.

Overall, our findings may support the rationale for adopting a TDM-guided approach of isavuconazole in the ICU scenario, especially when dealing with obese patients and/or with prolonged treatment, thus allowing the risk of undesired exposure potentially leading to failure or HTD to be reduced. An international position paper on antimicrobial TDM in critically ill patients reported routine isavuconazole TDM as neither recommended nor discouraged, based on linear and favorable PK properties [6]. Our findings, consistent with others previously reported [13,14,15,18], may suggest that some peculiar underlying pathophysiological alterations commonly reported in critically ill patients, by altering isavuconazole PK, may render the TDM-guided approach helpful. Specifically, obesity, on the one hand, by causing underexposure due to increased V_D_ [13,14,15,18] and hepatic insufficiency, on the other hand, by causing overexposure due to half-life prolongation [21,24,25] could be the two major areas of interest in applying TDM of isavuconazole. This is in agreement with a recent prospective study recommending the need for re-defining the role of a TDM-guided approach for antifungal therapy in critically ill patients [7]. Although applying a TDM-guided approach is mandatory for voriconazole due to the high interindividual pharmacokinetic variability [6,8], a paradigm shift to the need also for the other azole agents (namely, fluconazole, posaconazole, and isavuconazole) is currently occurring among critically ill patients. It is worth mentioning that, in the aforementioned prospective multicenter PK study, among 339 critically ill patients, the proportion of failure in attaining the desired PK/PD targets with azoles other than voriconazole ranged from 19.3% to 62.5% [7].

Limitations of our study should be acknowledged. First, the study was retrospective and monocentric, with a limited sample size. Second, multivariate analysis could not be performed due to the low occurrence of patients experiencing underexposure. This prevented us from identifying potential independent predictors of failure in attaining the desired isavuconazole exposure. Third, the limited small sample size may jeopardize the generalizability of our findings to larger groups of ICU patients. Fourth, the implication of multifactorial variables in the occurrence of HTD could not be ruled out due to the extremely complex case mix. Finally, other adverse events than HTD were not investigated.

In conclusion, our study suggested that implementing a real-time TDM-guided strategy for optimizing isavuconazole exposure could be a valuable tool for minimizing the risk of under- and/or overexposure in the ICU scenario, especially when dealing with obese patients or with prolonged treatment, as may occur in SOT recipients. Larger prospective studies would be warranted for confirming our findings.

## 4. Materials and Methods

### 4.1. Study Design

This study assessed retrospectively the potential role of a TDM-guided ECPA program of isavuconazole in preventing under- or overexposure in the ICU setting with the intent of improving efficacy and safety outcomes in critically ill patients. Selected patients were retrieved from those admitted to both the General and the Post-transplant ICUs of the IRCCS Azienda Ospedaliero-Universitaria of Bologna, Italy, in the period 1 March 2021–31 March 2025. All patients underwent at least one TDM-guided ECPA for optimizing exposure during antifungal prophylaxis or treatment with intravenous isavuconazole. Pediatrics and non-critically ill patients were excluded. A flowchart summarizing the study design is shown in Figure 4.

This study was approved by the Ethics Committee of IRCCS Azienda Ospedaliero-Universitaria of Bologna (n. 442/2021/Oss/AOUBo approved on 28 June 2021). Informed consent was waived due to the retrospective nature of the study.

### 4.2. Data Collection

For each patient, the following were retrieved: (a) demographic features [age, sex, weight, and body mass index (BMI)]; (b) clinical data [underlying diseases classified as solid organ transplant (SOT), hematological malignancies, or COVID-associated pulmonary aspergillosis (CAPA); need for mechanical ventilation, continuous renal replacement therapy (CRRT), extracorporeal membrane oxygenation (ECMO) or hemoadsorption treatment with cytosorb; vasopressor need; baseline partial pressure of oxygen to fraction of oxygen in the inhaled air ratio (PaO_2_/FiO_2_); and co-treatment with any strong CYP3A4 inhibitor and/or inducer]; (c) laboratory findings [serum albumin, serum aspartate aminotransferase (AST), serum alanine aminotransferase (ALT), total serum bilirubin, and serum gamma-glutamyltransferase (GGT)]; (d) microbiological data [IFI classification, serum and bronchoalveolar (BAL) galactomannan antigen (GM) index, and *Aspergillus* spp. clinical isolate]; (e) isavuconazole treatment data [daily posology, number of TDM-guided ECPA per patient, average isavuconazole trough concentrations (C_min_), and treatment duration]; (f) clinical/microbiological outcome data [hepatotoxicity, reduction in BAL GM index from baseline, occurrence of breakthrough IFI, and 30-day mortality rate].

CAPA was defined as COVID-19-positive patients admitted to the ICU with pulmonary infiltrates having at least one of the following criteria: serum GM index > 0.5 or BAL GM index > 1.0 or positive *Aspergillus* BAL culture or cavitating infiltrate (not attributable to other causes) in the area of the pulmonary infiltrate [30].

Hypoalbuminemia was defined as a serum albumin level < 3.5 g/dL, being classified as moderate or severe whenever <3.0 g/dL and <2.5 g/dL, respectively.

In patients requiring isavuconazole therapy, IFIs were classified as proven, probable, or possible according to the definition provided by the EORTC/MSGERC [31].

### 4.3. Outcome Definition

Reduction in BAL GM index from baseline, occurrence of breakthrough IFI, and 30-day mortality rate were selected as efficacy outcomes, whereas occurrence of hepatic test disturbance (HTD) was selected as safety outcome.

Reduction in BAL GM index on BAL was deemed as clinically relevant whenever a ≥50% decrease from baseline value was documented within 14 days.

Breakthrough IFI was defined as the occurrence of signs, symptoms, or findings of IFI coupled with the isolation of fungal specimen resistant to isavuconazole during isavuconazole prophylaxis or treatment, according to the latest definitions stated by the Mycoses Study Group Education and Research Consortium and European Confederation of Medical Mycology [32].

HTD was assessed and categorized according to the Common Terminology Criteria for Adverse Events v. 5.0 (CTCAE) [33]. Baseline hepatic tests were defined as normal whenever all of the hepatic biochemical parameters were within the normal range before starting isavuconazole treatment or as altered whenever one or more of the hepatic biochemical parameters were above the upper normal threshold (i.e., serum AST or ALT > 50 U/L, serum GGT > 55 U/L, or serum total bilirubin > 1.2 g/dL).

### 4.4. Isavuconazole Dosing Regimens, Sampling Procedure, and Definition of Optimal Exposure

Treatment with isavuconazole was started according to the dosing regimen recommended in the product summary (intravenous loading dose of 200 mg every 8 h for the first 48 h followed by an intravenous maintenance dose (MD) of 200 mg every 24 h). Isavuconazole dosing adjustments were performed according to a real-time TDM-guided ECPA program, as previously reported [34].

Blood samples for assessing isavuconazole C_min_ were collected 5–15 min before daily dosing after at least 72 h from starting treatment and eventually recollected every 3–5 days whenever feasible. After centrifugation and plasma separation, isavuconazole concentrations in plasma were measured by means of a liquid chromatography–tandem mass spectrometry (LC–MS/MS) commercially available method (Chromsystems Instruments and Chemicals GmbH, Munich, Germany) [35].

The desired isavuconazole C_min_ was set between 1.0 and 5.1 mg/L according to recent evidence suggesting that maintaining C_min_ within this range was associated with a good risk–benefit ratio [14,20,34]. Based on the profile of isavuconazole exposure during the overall treatment duration, three different scenarios were defined: (a) adequate exposure, defined as C_min_ always within the desired range during the entire treatment period; (b) transient or persistent overexposure, defined as C_min_ being above the desired range not in all or in all TDM instances, respectively; (c) transient or persistent underexposure, defined as C_min_ being below the desired range not in all or in all TDM instances, respectively.

### 4.5. Statistical Analysis

Demographics and clinical characteristics of included patients were summarized by using absolute frequencies and percentages for categorical variables and median with interquartile ranges (IQR) for continuous variables. Univariate analyses were performed [by means of the Fisher’s exact test or the chi-squared test (for categorical variables) or the Mann–Whitney U test (for continuous variables)] either for identifying potential variables associated with isavuconazole under- or overexposure or for comparing features in SOT vs. non-SOT critically ill patients. Likelihood of adequate isavuconazole C_min_ attainment was assessed in relation to the elapsed time from starting therapy by calculating the proportion of isavuconazole underexposure, desired exposure, and overexposure at each of three selected timepoints, namely <7, 7–28, and >28 days. Statistical significance was further evaluated by means of the chi-square. The relationship between isavuconazole exposure and HTD occurrence was also investigated in a descriptive fashion by splitting patients with HTD occurrence into three groups according to isavuconazole exposure over time (i.e., patients having always desired exposure; patients with transient or persistent overexposure; and patients with transient or persistent underexposure). Each group was successively divided into two subgroups based on having or not altered baseline hepatic biochemical parameters.

Statistical analysis was performed by using MedCalc for Windows (MedCalc statistical software, version 19.6.1, MedCalc Software Ltd., Ostend, Belgium). *p* values < 0.05 were considered statistically significant.

## Figures and Tables

**Figure 1 antibiotics-14-00777-f001:**
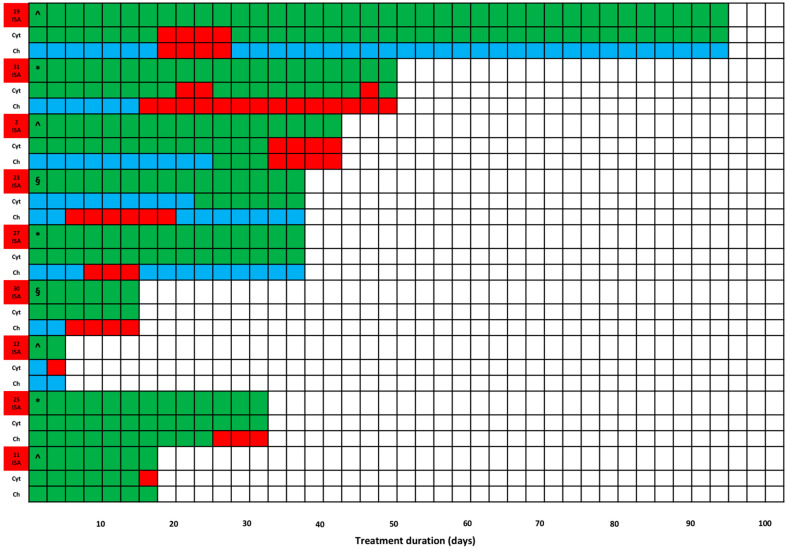
Trends of HTD over time among patients always having desired isavuconazole exposure (*n* = 9), stratified by having (*n* = 7) or not (*n* = 2) altered baseline hepatic biochemical parameters. For each patient, there are three rows indicating, in the upper, the type of isavuconazole exposure (ID patient; green, desired); in the intermediate, the type of serum AST/ALT levels (Cyt) (green, normal; light blue, baseline altered; red, increase according to the CTCAE 5.0 criteria); in the lower, the type of serum GGT/total bilirubin levels (Ch) (green, normal; light blue, baseline altered; red, increase according to the CTCAE 5.0 criteria). Symbols: * non-hepatic SOT recipients; § liver transplant recipients; ^ non-SOT ICU patients.

**Figure 2 antibiotics-14-00777-f002:**
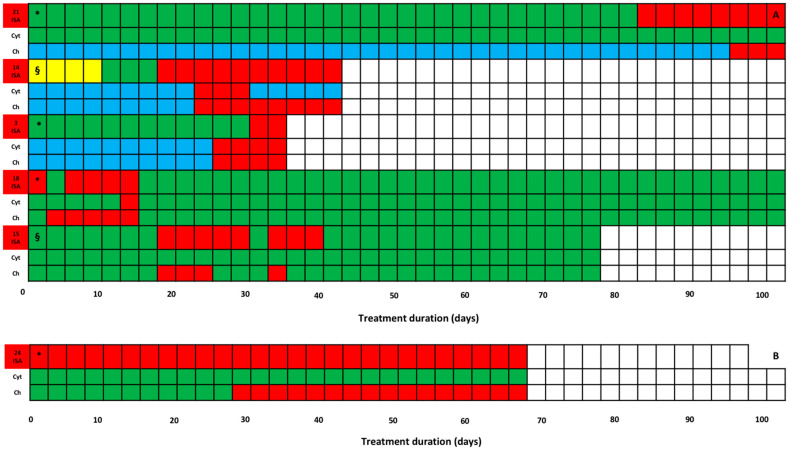
Trends of HTD over time among the nine patients experiencing HTD while having transient ((**A**), *n* = 5) or persistent ((**B**), *n* = 1) isavuconazole overexposure, stratified by having (*n* = 3) or not (*n* = 3) altered baseline hepatic biochemical parameters. For each patient, there are three rows indicating, in the upper, the type of isavuconazole exposure (green, desired; red, overexposure; yellow, underexposure); in the intermediate, the type of serum AST/ALT levels (Cyt) (green, normal; light blue, baseline altered; red, increase according to the CTCAE 5.0 criteria); in the lower, the type of serum GGT/total bilirubin levels (Ch) (green, normal; light blue, baseline altered; red, increase according to the CTCAE 5.0 criteria). Symbols: * non-hepatic SOT recipients; § liver transplant recipients.

**Figure 3 antibiotics-14-00777-f003:**
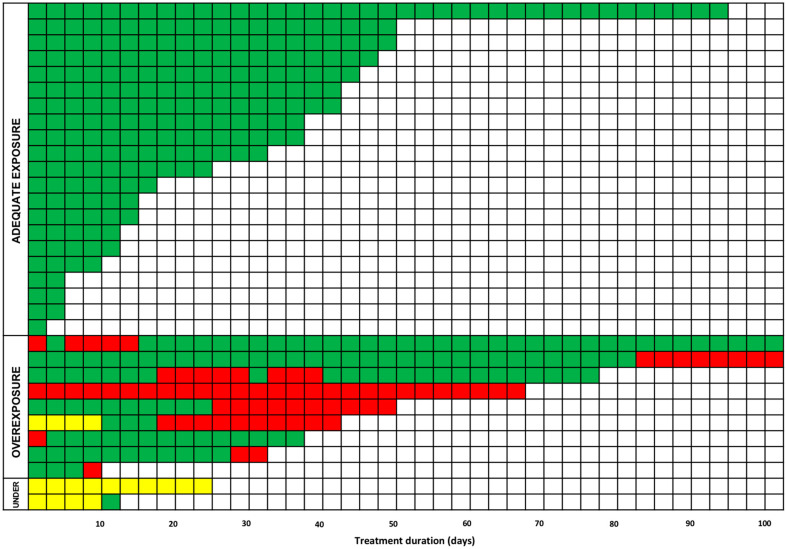
Trends of isavuconazole exposure over time in all of the included patients (*n* = 32). Type of isavuconazole exposure: green, desired; red, overexposure; yellow, underexposure.

**Figure 4 antibiotics-14-00777-f004:**
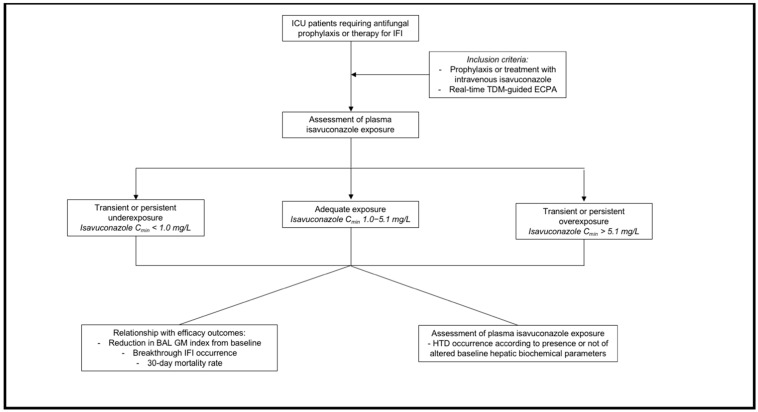
Flowchart of the study design.

**Table 1 antibiotics-14-00777-t001:** Demographics and clinical characteristics of ICU patients receiving intravenous isavuconazole.

Variables	Overall Patients (*n* = 32)
Demographics	
Age (yrs; Median; [IQR])	62.0 (51.0–65.0)
Gender (male/female; *n* [%])	25/7 (78.1/21.9)
Weight (kg; Median; [IQR])	73.0 (60.8–85.8)
Body mass index (kg/m^2^; Median; [IQR])	25.5 (22.3–27.8)
Obesity (*n*; [%])	5 (15.6)
Underlying disease (*n*; [%])	
Solid organ transplant recipient	21 (65.6)
Hematological malignancies	6 (18.8)
CAPA	5 (15.6)
Pathophysiological conditions	
SOFA score (Median; [IQR])	8 (5–11)
Vasopressors (*n*; [%])	19 (59.4)
Mechanical ventilation (*n*; [%])	25 (78.1)
PaO_2_/FiO_2_ ratio (Median; [IQR])	166.0 (122.8–197.3)
PaO_2_/FiO_2_ ratio < 200 (*n*; [%])	24 (75.0)
Continuous renal replacement therapy (*n*; [%])	16 (50.0)
Cytosorb hemoadsorption (*n*; [%])	2 (6.3)
Extracorporeal membrane oxygenation (*n*; [%])	0 (0.0)
Serum albumin (g/dL; Median; [IQR])	2.8 (2.5–3.1)
Patients with moderate/severe hypoalbuminemia (*n*; [%])	22 (68.7)
Concomitant use of CYP3A4 strong inducers * (*n*; [%])	1 (3.1)
Concomitant use of CYP3A4 strong inhibitors * (*n*; [%])	0 (0.0)
IFI classification ** (*n*; [%])	
Proven	1 (3.1)
Probable	25 (78.1)
Possible	2 (6.3)
No IFI	4 (12.5)
Documentation of IFI	
Patients with *Aspergillus* spp. isolated from BAL (*n*; [%])	6 (18.7)
Patients with positive galactomannan antigen on BAL (*n*; [%])	28 (87.5)
Patients with positive galactomannan antigen on serum (*n*; [%])	3 (9.4)
Galactomannan antigen index on BAL in positive patients (Median; [IQR])	2.6 (2.0–4.2)
Galactomannan antigen index on serum in positive patients (Median; [IQR])	2.9 (2.0–7.4)
Type of isavuconazole treatment (*n*; [%])	
First-line therapy	6 (18.7)
Second-line therapy	22 (68.7)
First-line prophylaxis	2 (6.3)
Second-line prophylaxis	2 (6.3)
Isavuconazole treatment regimens and PK/PD target attainment	
Isavuconazole MD (mg/die; Median; [IQR])	200 (200–200)
Treatment duration (days; Median; [IQR])	35.0 (12.5–47.75)
Overall TDM-guided ECPA	166
TDM-guided ECPA per patient (Median; [IQR])	4 (2–7)
Average isavuconazole C_min_ (mg/L; Median; [IQR])	3.5 (2.1–4.6)
(patients; *n*; [%])	29 (90.6)
Below desired range (patients; *n*; [%])	2 (6.3)
Above desired range (patients; *n*; [%])	1 (3.1)
Within desired range (ECPA; *n*; [%])	131 (78.9)
Below desired range (ECPA; *n*; [%])	7 (4.2)
Above desired range (ECPA; *n*; [%])	28 (16.9)
Outcome (*n*; [%])	
HTD occurrence with no baseline alteration (*n*; [%])	5 (15.6)
Reduction ≥50% from baseline of galactomannan antigen on BAL within 14 days ***	16/23 (69.6)
Breakthrough IFI	0 (0.0)
30-day mortality rate	10 (31.3)

BAL: bronchoalveolar lavage; CAPA: COVID-associated pulmonary aspergillosis; C_min_: trough concentration; ECPA: expert clinical pharmacological advice; HTD: hepatic test disturbance; ICU: intensive care unit; IFI: invasive fungal infection; IQR: interquartile range; MD: maintenance dose; PaO_2_/FiO_2_: partial pressure of oxygen to fraction of oxygen in the inhaled air ratio; PK/PD: pharmacokinetic/pharmacodynamic; SOFA: sequential organ failure assessment; TDM: therapeutic drug monitoring. * According to the DrugBank list of CYP3A4 strong inducers and inhibitors [16,17]. ** According to definition provided by EORTC/MSGERC. *** Follow-up galactomannan antigen on BAL was available in 23/28 of patients with positive value at baseline.

**Table 2 antibiotics-14-00777-t002:** Univariate analysis comparing patients with isavuconazole trough concentrations below desired range and those with no isavuconazole trough concentrations below desired range.

Variables	Patients with No Isavuconazole C_min_ Below Desired Range (*n* = 30)	Patients with Isavuconazole C_min_ Below Desired Range (*n* = 2)	*p* Value
Demographics			
Age (yrs; Median; [IQR])	62.5 (51.0–65.0)	53.0 (50.5–55.5)	0.21
Gender (male/female; *n* [%])	23/7 (76.7/23.3)	2/0 (100.0/0.0)	0.99
Weight (kg; Median; [IQR])	72.5 (60.3–85.0)	95.5 (83.3–107.8)	0.24
Body mass index (kg/m^2^; Median; [IQR])	25.5 (22.0–27.7)	30.1 (26.6–33.6)	**<0.001**
Obesity (*n*; [%])	4 (13.3)	1 (50.0)	0.29
Underlying disease (*n*; [%])			
Solid organ transplant recipient	21 (70.0)	0 (0.0)	0.11
Hematological malignancies	5 (16.7)	1 (50.0)	0.34
CAPA	4 (13.3)	1 (50.0)	0.29
Pathophysiological conditions			
SOFA score (Median; [IQR])	8.5 (5.25–11.0)	6.5 (5.75–7.25)	0.48
Vasopressors (*n*; [%])	17 (56.7)	2 (100.0)	0.50
Mechanical ventilation (*n*; [%])	23 (76.7)	2 (100.0)	0.99
PaO_2_/FiO_2_ ratio (Median; [IQR])	166.0 (122.25–195.5)	181.5 (171.75–191.25)	0.56
PaO_2_/FiO_2_ ratio < 200 (*n*; [%])	23 (76.7)	1 (50.0)	0.44
Continuous renal replacement therapy (*n*; [%])	15 (50.0)	1 (50.0)	0.99
Use of cytosorb hemoadsorption (*n*; [%])	1 (3.3)	1 (50.0)	0.12
Extracorporeal membrane oxygenation (*n*; [%])	0 (0.0)	0 (0.0)	0.99
Serum albumin (g/dL; Median; [IQR])	2.9 (2.6–3.1)	2.4 (2.2–2.5)	0.15
Patients with moderate/severe hypoalbuminemia (*n*; [%])	20 (66.7)	2 (100.0)	0.99
Concomitant use of CYP3A4 strong inducers * (*n*; [%])	0 (0.0)	1 (50.0)	0.06
IFI classification ** (*n*; [%])			
Proven	1 (3.3)	0 (0.0)	0.99
Probable	24 (80.0)	1 (50.0)	0.40
Possible	2 (6.7)	0 (0.0)	0.99
No IFI	3 (10.0)	1 (50.0)	0.24
Type of isavuconazole treatment (*n*; [%])			
First-line therapy	6 (20.0)	0 (0.0)	0.99
Subsequent line of therapy	21 (70.0)	1 (50.0)	0.53
First-line prophylaxis	1 (3.3)	1 (50.0)	0.12
Second-line prophylaxis	2 (6.7)	0 (0.0)	0.99
Isavuconazole treatment regimens and outcome			
Isavuconazole MD (mg/die; Median; [IQR])	200 (200–200)	250 (200–300)	**<0.001**
Treatment duration (days; Median; [IQR])	37.0 (13.25–49.25)	19.0 (15.0–23.0)	0.37
HTD occurrence with no baseline alteration (*n*; [%])	5 (16.7)	0 (0.0)	0.99
Reduction ≥ 50% from baseline of galactomannan antigen on BAL within 14 days *** (*n*; [%])	15/21 (71.4)	1/2 (50.0)	0.53
Breakthrough IFI (*n*; [%])	0 (0.0)	0 (0.0)	0.99
30-day mortality rate (*n*; [%])	9 (30.0)	1 (50.0)	0.53

BAL: bronchoalveolar lavage; CAPA: COVID-associated pulmonary aspergillosis; HTD: hepatic test disturbance; IFI: invasive fungal infection; IQR: interquartile range; MD: maintenance dose; PaO_2_/FiO_2_: partial pressure of oxygen to fraction of oxygen in the inhaled air ratio; SOFA: sequential organ failure assessment. * According to the DrugBank list of CYP3A4 strong inducers and inhibitors [16,17]. ** According to definition provided by EORTC/MSGERC. *** Follow-up galactomannan antigen on BAL was available in 23/28 of patients with positive value at baseline. Bold represents variables having statistically significance.

**Table 3 antibiotics-14-00777-t003:** Number of isavuconazole C_min_ being at different timepoint instances below, within, or above the desired range.

Isavuconazole C_min_ (mg/L)	Day < 7	Day 7–28	Day > 28	*p* Value
<1.0 (*n*; [%])	4/38 (10.5%)	3/70 (4.3%)	0/58 (0.0%)	**<0.001**
1.0–5.1 (*n*; [%])	32/38 (84.2%)	58/70 (82.9%)	41/58 (70.7%)	
>5.1 (*n*; [%])	2/38 (5.3%)	9/70 (12.8%)	17/58 (29.3%)	

C_min_: trough concentrations. Bold represents variables having statistically significance.

**Table 4 antibiotics-14-00777-t004:** Univariate analysis comparing features in SOT vs. non-SOT critically ill patients receiving IV isavuconazole.

Variables	SOT Patients (*n* = 21)	Non-SOT Patients (*n* = 11)	*p* Value
Demographics			
Age (yrs; Median; [IQR])	62.0 (51.0–64.0)	64.0 (54.5–65.0)	0.55
Gender (male/female; *n* [%])	17/4 (81.0/19.0)	8/3 (72.7/27.3)	0.67
Weight (kg; Median; [IQR])	68.5 (60.0–80.0)	84.0 (68.0–100.0)	0.10
Body mass index (kg/m^2^; Median; [IQR])	24.2 (21.1–26.6)	27.4 (23.5–30.4)	0.10
Obesity (*n*; [%])	2 (9.5)	3 (27.3)	0.31
Pathophysiological conditions			
SOFA score (Median; [IQR])	8.0 (5.0–10.0)	8.0 (6.5–11.0)	0.56
Vasopressors (*n*; [%])	12 (57.1)	7 (63.6)	0.99
Mechanical ventilation (*n*; [%])	16 (76.2)	9 (81.8)	0.99
PaO_2_/FiO_2_ ratio (Median; [IQR])	180.0 (135.0–209.0)	136.0 (91.5–179.0)	0.10
PaO_2_/FiO_2_ ratio < 200 (*n*; [%])	15 (71.4)	9 (81.8)	0.68
Continuous renal replacement therapy (*n*; [%])	12 (57.1)	4 (36.4)	0.46
Use of cytosorb hemoadsorption (*n*; [%])	1 (4.8)	1 (9.1)	0.99
Extracorporeal membrane oxygenation (*n*; [%])	0 (0.0)	0 (0.0)	0.99
Serum albumin (g/dL; Median; [IQR])	2.88 (2.63–3.11)	2.73 (2.41–3.22)	0.76
Patients with moderate/severe hypoalbuminemia (*n*; [%])	14 (66.7)	8 (72.7)	0.99
Concomitant use of CYP3A4 strong inducers * (*n*; [%])	0 (0.0)	1 (9.1)	0.34
IFI classification ** (*n*; [%])			
Proven	1 (4.8)	0 (0.0)	0.99
Probable	18 (85.7)	7 (63.6)	0.20
Possible	2 (9.5)	0 (0.0)	0.53
Prophylaxis	0 (0.0)	4 (36.4)	**0.009**
Type of isavuconazole treatment (*n*; [%])			
First-line therapy	4 (19.0)	2 (18.2)	0.99
Subsequent line of therapy	17 (81.0)	5 (45.4)	0.06
Prophylaxis	0 (0.0)	4 (36.4)	**0.009**
Isavuconazole treatment regimens and outcome			
Isavuconazole MD (mg/die; Median; [IQR])	200 (200–200)	200 (200–200)	0.33
Treatment duration (days; Median; [IQR])	42 (31–50)	13 (8–26)	**0.02**
Average isavuconazole C_min_ (mg/L; Median; [IQR])	3.44 (2.41–4.03)	2.20 (1.36–2.70)	**0.006**
No. of TDM with isavuconazole underexposure *** (*n*; [%])	2/137 (1.5)	5/29 (17.2)	**0.002**
No. of TDM with isavuconazole overexposure *** (*n*; [%])	27/137 (19.7)	0/29 (0.0)	**0.005**
HTD occurrence in pts with normal baseline values (*n*; [%])	4 (19.0)	1 (9.1)	0.64
Reduction ≥ 50% from baseline of galactomannan antigen on BAL within 14 days **** (*n*; [%])	10/16 (62.5)	6/7 (85.7)	0.37
Breakthrough IFI (*n*; [%])	0 (0.0)	0 (0.0)	0.99
30-day mortality rate (*n*; [%])	5 (23.8)	5 (45.5)	0.21

BAL: bronchoalveolar lavage; HTD: hepatic test disturbance: IFI: invasive fungal infection; IQR: interquartile range; MD: maintenance dose; PaO_2_/FiO_2_: partial pressure of oxygen to fraction of oxygen in the inhaled air ratio; SOFA: sequential organ failure assessment; SOT: solid organ transplant; TDM: therapeutic drug monitoring. * According to the DrugBank list of CYP3A4 strong inducers and inhibitors [16,17]. ** According to definition provided by EORTC/MSGERC. *** A total of 137 and 29 TDM instances were performed in SOT and non-SOT patients, respectively. **** Follow-up galactomannan antigen on BAL was available in 23/28 of patients with positive value at baseline. Bold represents variables having statistically significance.

## Data Availability

The data presented in this study are available on request from the corresponding author. The data are not publicly available due to privacy concerns.

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
