# Peer review of "TDM-Based Approach for Properly Managing Intravenous Isavuconazole Treatment in a Complex Case Mix of Critically Ill Patients"

_antibiotics, 2025, doi:10.3390/antibiotics14080777_

Round 1
Reviewer 1 Report
Comments and Suggestions for Authors
Comment:
This article provides valuable insights for the healthcare team on the application of therapeutic drug monitoring (TDM) in antifungal therapy. However, several aspects could be improved to enhance the clarity, depth, and impact of the manuscript:
- Please shorten or concise the Title name while retain the main idea and finding or outcome.
- The abstract should be rewritten to improve clarity and completeness. In particular, the methodology section requires more detailed and coherent description.
- Introduction; please include the review literature on physicochemical property of isavuconazole with its pharmacokinetic and pharmacodynamic especially volume of distribution and metabolized product.
- Please cite supporting ref for “Additionally, voriconazole plasma exposure may be unpredictable 64 due to the genetic polymorphism of CYP2C19 causing wide interindividual variability in 65 biotransformation, thus making therapeutic drug monitoring (TDM) mandatory nowa-66 days.” In Introduction.
- The inclusion and exclusion criteria should be clearly stated and expanded upon.
- The reported outcomes would benefit from being categorized (e.g., efficacy vs. safety) to aid reader comprehension.
- The statistical methods used to assess relationships should be explicitly described in more detail with cited references.
- The figures are currently difficult to interpret. Reformatting or clearer labeling or style of graph presentation may improve understanding.
- Regarding the association between low isavuconazole exposure and BMI, additional explanation should be provided, particularly concerning the drug’s pharmacokinetic properties that may underlie this finding.
- Incorporating more discussion on the pharmacokinetics and pharmacodynamics of isavuconazole is required to strengthen the rationale for TDM in clinical practice with more supporting refs.
- Expanding the discussion to include the broader role of TDM in antifungal therapy could enhance the article’s appeal and relevance to a wider readership or comparison with the other previous studies or related articles.
- The overall writing quality should be refined to improve readability and engagement for the target audience. The schematic diagram or flow chart should be created and applied for more understanding to the readers especially for methodology or before conclusion part.
Author Response
RESPONSE TO REVIEWERS
Manuscript ID: antibiotics-3729095 “Role of a TDM-based approach in properly managing efficacy and/or safety of intravenous isavuconazole treatment in a highly complex case mix of critically ill patients” by Gatti et al.
Dear Editor,
We would like to thank you for the opportunity to resubmit a revised version of this manuscript. We appreciated the reviewer’s constructive comments. All have been carefully considered and incorporated, where and whenever possible, in the revision. Furthermore, as suggested we carefully reviewed the English language in order to improve the readability.
Our point-by-point responses are provided below.
Q= QUERY; A= ANSWER
Reviewer #1:
This article provides valuable insights for the healthcare team on the application of therapeutic drug monitoring (TDM) in antifungal therapy. However, several aspects could be improved to enhance the clarity, depth, and impact of the manuscript:
We thank the reviewer for appreciating our paper.
Q1. Please shorten or concise the Title name while retain the main idea and finding or outcome.
A1. We thank the reviewer for this comment. We shortened the title of the manuscript as per required (“TDM-based approach for properly managing intravenous isavuconazole treatment in a complex case mix of critically ill patients”).
Q2. The abstract should be rewritten to improve clarity and completeness. In particular, the methodology section requires more detailed and coherent description.
A2. We thank the reviewer for this comment. We revised the abstract as suggested, particularly the section concerning the methods of the study, also according to comment No. 6 and No. 7.
Q3. Introduction; please include the review literature on physicochemical property of isavuconazole with its pharmacokinetic and pharmacodynamic especially volume of distribution and metabolized product.
A3. Thank you for this suggestion. We included data on physicochemical, pharmacokinetic, and pharmacodynamic properties of isavuconazole in the Introduction section (refer to Line 72-73).
Q4. Please cite supporting ref for “Additionally, voriconazole plasma exposure may be unpredictable 64 due to the genetic polymorphism of CYP2C19 causing wide interindividual variability in 65 biotransformation, thus making therapeutic drug monitoring (TDM) mandatory nowa-66 days.” In Introduction.
A4. Thank you for this suggestion. We added a specific reference supporting the unpredictability of voriconazole plasma exposure and the consequent need for the implementation of a TDM-guided approach.
Q5. The inclusion and exclusion criteria should be clearly stated and expanded upon.
A5. We thank the reviewer for this comment, allowing us to better clarify the selected inclusion and exclusion criteria in the Methods section (refer to Line 340-345).
Q6. The reported outcomes would benefit from being categorized (e.g., efficacy vs. safety) to aid reader comprehension.
A6. We thank the reviewer for this useful suggestion. We added a specific section concerning outcome definition in the Methods section (refer to Line 376-382), in order to improve the clarity and comprehension of this relevant issue.
Q7. The statistical methods used to assess relationships should be explicitly described in more detail with cited references.
A7. We thank the reviewer for this comment. We detailed the statistical analyses performed for assessing the relationship between isavuconazole exposure at the different timepoints, as well as those between isavuconazole exposure and HTD occurrence as suggested (refer to Methods section; Line 423-432).
Q8. The figures are currently difficult to interpret. Reformatting or clearer labeling or style of graph presentation may improve understanding.
A8. We thank the reviewer for this comment. As also suggested by Reviewer 2 in the first comment, we modified the Figure 1 legend in order to improve readability, removing also the description of the colors yellow and red considering that no case of underexposure and/or overexposure was reported among the presented nine patients, in order to minimize the risk of misunderstanding for the readers. Furthermore, we improved the quality of the figures for a better clarity.
Q9. Regarding the association between low isavuconazole exposure and BMI, additional explanation should be provided, particularly concerning the drug’s pharmacokinetic properties that may underlie this finding.
A9. We thank the reviewer for this comment, allowing us to better discuss this important issue concerning the relationship between isavuconazole underexposure and BMI. We added additional explanation in the Discussion section (refer to Line 243-248) as required.
Q10. Incorporating more discussion on the pharmacokinetics and pharmacodynamics of isavuconazole is required to strengthen the rationale for TDM in clinical practice with more supporting refs.
A10. We thank the reviewer for this relevant comment, allowing us to better discuss and strengthen the rationale for performing isavuconazole TDM in clinical practice. We added this issue in the Discussion section (refer to Line 302-319).
Q11. Expanding the discussion to include the broader role of TDM in antifungal therapy could enhance the article’s appeal and relevance to a wider readership or comparison with the other previous studies or related articles.
A11. We thank the reviewer for this useful suggestion. We expanded the Discussion section (refer to Line 302-319) by discussing our findings in the context of the broader role of TDM implementation in antifungal therapy in ICU scenario.
Q12. The overall writing quality should be refined to improve readability and engagement for the target audience. The schematic diagram or flow chart should be created and applied for more understanding to the readers especially for methodology or before conclusion part.
A12. We thank the reviewer for this suggestion. We extensively revised our manuscript for improving the readability as suggested. English language was extensively revised and improved. A dedicated flow chart has been added to the Methods section (refer to Figure 4) for improving the understanding of the study design.
Reviewer 2 Report
Comments and Suggestions for Authors
This article presents an interesting and clinically relevant study on optimising isavuconazole dosing in critically ill patients. However, after reviewing the manuscript, the authors should address several issues.
First, Figure 1's title should be modified to more appropriately convey its meaning. The title should say that the figure shows patients with HTD who have different exposure profiles, such as overexposure (for example, "Trends of HTD over time among patients experiencing HTD while having varied isavuconazole exposure, including overexposure (n = 9) …"). The authors should also think about taking out the description of the color yellow ("yellow, underexposure") from the legend because it doesn't show up in this figure.
Second, Figure 2A shows a discrepancy between the number of patients given in the description (n = 8) and the number of patients depicted in the graph (n = 5).
The authors rightly point out that the small sample size (n = 32), particularly the very small sizes of the subgroups (e.g. n = 2 patients with underexposure in Table 2 and n = 11 non-SOT patients in Table 4), is a major flaw of this study and made multivariate analysis impossible. However, the discussion should emphasize more clearly how this small scale may affect the interpretation of observed 'trends' or the lack of statistical significance, which limits the applicability of the conclusions to a larger group of critically ill patients.
Minor comments:
line 57, the scientific name "Aspergillus spp." should be italicized
Line 208: typo. In the introductory text of Table 4, '17.2 vs. 1.5%' appears.
In Table 4, the lines 'No. of TDM with isavuconazole underexposure (n; [%])' for SOT and non-SOT patients use the formats '2/137 (1.5)' and '5/29 (17.2)'. While this is clear, it is inconsistent with the general 'n ([%])' format used in tables for categorical variables. It would be better to standardise the formatting.
Comments on the Quality of English LanguageThe manuscript should be reviewed to ensure it is accurate and fluent in English.
Author Response
RESPONSE TO REVIEWERS
Manuscript ID: antibiotics-3729095 “Role of a TDM-based approach in properly managing efficacy and/or safety of intravenous isavuconazole treatment in a highly complex case mix of critically ill patients” by Gatti et al.
Dear Editor,
We would like to thank you for the opportunity to resubmit a revised version of this manuscript. We appreciated the reviewer’s constructive comments. All have been carefully considered and incorporated, where and whenever possible, in the revision. Furthermore, as suggested we carefully reviewed the English language in order to improve the readability.
Our point-by-point responses are provided below.
Q= QUERY; A= ANSWER
Reviewer #2:
This article presents an interesting and clinically relevant study on optimising isavuconazole dosing in critically ill patients. However, after reviewing the manuscript, the authors should address several issues.
We thank the reviewer for appreciating our paper.
Q1. First, Figure 1's title should be modified to more appropriately convey its meaning. The title should say that the figure shows patients with HTD who have different exposure profiles, such as overexposure (for example, "Trends of HTD over time among patients experiencing HTD while having varied isavuconazole exposure, including overexposure (n = 9) …"). The authors should also think about taking out the description of the color yellow ("yellow, underexposure") from the legend because it doesn't show up in this figure.
A1. We thank the reviewer for this comment, allowing us to better clarify the meaning of Figure 1. In this figure, only the nine patients having always desired isavuconazole exposure (i.e., an isavuconazole Cmin ranging from 1 mg/L to 5.1 mg/L in all TDM assessments) who developed HTD were showed. We modified the figure legend in “Trends of HTD over time among patients having always desired isavuconazole exposure (n=9) and stratified by having (n=7) or not (n=2) altered baseline hepatic biochemical parameters”. Furthermore, we removed the description of the colors yellow and red from Figure 1 legend considering that no case of underexposure and/or overexposure was reported among the presented nine patients, in order to minimize the risk of misunderstanding for the readers.
Q2. Second, Figure 2A shows a discrepancy between the number of patients given in the description (n = 8) and the number of patients depicted in the graph (n = 5).
A2. We thank the reviewer for this suggestion. We corrected the discrepancy existing between the Figure 2A and the specific figure legend.
Q3. The authors rightly point out that the small sample size (n = 32), particularly the very small sizes of the subgroups (e.g. n = 2 patients with underexposure in Table 2 and n = 11 non-SOT patients in Table 4), is a major flaw of this study and made multivariate analysis impossible. However, the discussion should emphasize more clearly how this small scale may affect the interpretation of observed 'trends' or the lack of statistical significance, which limits the applicability of the conclusions to a larger group of critically ill patients.
A3. We thank the reviewer for this useful comment. We emphasize this limitation in the Discussion section (refer to Line 322-325).
Minor comments:
Q4. line 57, the scientific name "Aspergillus spp." should be italicized
A4. Thank you for this suggestion. We italicized the name as required (refer to Line 60)
Q5. Line 208: typo. In the introductory text of Table 4, '17.2 vs. 1.5%' appears.
A5. Thank you for this suggestion. We corrected the typo as required (refer to Line 208).
Q6. In Table 4, the lines 'No. of TDM with isavuconazole underexposure (n; [%])' for SOT and non-SOT patients use the formats '2/137 (1.5)' and '5/29 (17.2)'. While this is clear, it is inconsistent with the general 'n ([%])' format used in tables for categorical variables. It would be better to standardise the formatting.
A6. Thank you for this suggestion. In order to maintain consistency with the general format, we modified accordingly and we added a specific footnote reporting the overall number of TDM performed in each group.
Finally, English language was extensively revised and improved.
Round 2
Reviewer 1 Report
Comments and Suggestions for Authors
comment:
The chemical structure of isavuconazole and physicochemical properties (molecular weight, solubility, pKa, logK, and so on) of isavuconazole have to add and address in Introduction with supporting references.
The review on antifungal activities both on in vitro and in vivo studies (MIC, MBC or other vulues) of isavuconazole have to include in Introduction.
Please indicate what is the meaning of value in (......) in last column of Table 1 and column 2,3 of Table 2 and column 2-4 of Table 3 and column 2-3 of Table 4.
Please move the legend of Figure 1. to the beneath part of figure and have the footnote for the abbreviation of its first column.
Line 72 please specific the value of protein binding no need i.e Line 245-246 no need i.e.
Line 320 which in the discussion about the limitation, please mention by using the number (first, second , third..) of using the conjunction words to make more understanding and make more influence content.
Author Response
RESPONSE TO REVIEWERS
Manuscript ID: antibiotics-3729095 “Role of a TDM-based approach in properly managing efficacy and/or safety of intravenous isavuconazole treatment in a highly complex case mix of critically ill patients” by Gatti et al.
Dear Editor,
We would like to thank you for the opportunity to resubmit a novel revised version of this manuscript. We appreciated the reviewer’s constructive comments. All have been carefully considered and incorporated in the revision. Modified or added parts are highlighted in green.
Our point-by-point responses are provided below.
Q= QUERY; A= ANSWER
Reviewer #1:
Q1. The chemical structure of isavuconazole and physicochemical properties (molecular weight, solubility, pKa, logK, and so on) of isavuconazole have to add and address in Introduction with supporting references.
A1. We thank the reviewer for this comment. We added in the Introduction section (Line 72-74) the suggested data.
Q2. The review on antifungal activities both on in vitro and in vivo studies (MIC, MBC or other vulues) of isavuconazole have to include in Introduction.
A2. We thank the reviewer for this comment. We added in the Introduction section (Line 77-87) the suggested data.
Q3. Please indicate what is the meaning of value in (......) in last column of Table 1 and column 2,3 of Table 2 and column 2-4 of Table 3 and column 2-3 of Table 4.
A3. Thank you for this comment, allowing us to better clarify this issue. For Table 1, 2 and 4, the meaning of the values reported in brackets is reported in the first column (for example: Age (yrs; Median; [IQR]) means that the values reported in column 2 and 3 are median age in years and interquartile range in brackets). For column 2-4 of Table 3 we added this information in the first column of the table.
Q4. Please move the legend of Figure 1. to the beneath part of figure and have the footnote for the abbreviation of its first column.
A4. We thank the reviewer for this suggestion. We moved the legend of Figure 1 below the figure as suggested. Furthermore, we also added a footnote for the abbreviation reported in the first column.
Q5. Line 72 please specific the value of protein binding no need i.e Line 245-246 no need i.e.
A5. Thank you for this suggestion. We removed the term “i.e.” as suggested. Furthermore, the value of protein binding for isavuconazole is 98-99% as specified in the Introduction section (refer to Line 74).
Q6. Line 320 which in the discussion about the limitation, please mention by using the number (first, second, third..) of using the conjunction words to make more understanding and make more influence content.
A6. Thank you for this suggestion. We mentioned each limitation by using the number in order to improve the readability of this section (refer to Line 333-341).
Reviewer 2 Report
Comments and Suggestions for Authors
Thank you for the revised manuscript. I have no further comments.
Author Response
RESPONSE TO REVIEWERS
Manuscript ID: antibiotics-3729095 “Role of a TDM-based approach in properly managing efficacy and/or safety of intravenous isavuconazole treatment in a highly complex case mix of critically ill patients” by Gatti et al.
Dear Editor,
We would like to thank you for the opportunity to resubmit a novel revised version of this manuscript. We appreciated the reviewer’s constructive comments. All have been carefully considered and incorporated in the revision. Modified or added parts are highlighted in green.
Our point-by-point responses are provided below.
Q= QUERY; A= ANSWER
Reviewer #2:
Thank you for the revised manuscript. I have no further comments.
We thank the reviewer for appreciating the revised version of our manuscript.
Round 3
Reviewer 1 Report
Comments and Suggestions for Authors
Now the authors well improve the content.
Comments on the Quality of English LanguageThe improvement is conducted from the authors.